# A Scoping Review and Conceptual Model of Social Participation and Mental Health among Refugees and Asylum Seekers

**DOI:** 10.3390/ijerph16204027

**Published:** 2019-10-21

**Authors:** Maria Niemi, Hélio Manhica, David Gunnarsson, Göran Ståhle, Sofia Larsson, Fredrik Saboonchi

**Affiliations:** 1Department of Public Health Science, Karolinska Institutet, Solnavägen 1E, 104 31 Stockholm, Sweden; helio.manhica@ki.se; 2Center for Social Sustainability, Department of Neurobiology, Care Sciences and Society, Karolinska Institutet, Alfred Nobels Allé 23, 141 52 Huddinge, Sweden; 3School of Historical and Contemporary Studies, Södertörn University, Alfred Nobels Allée 7, 141 89 Huddinge, Sweden; david.gunnarsson@sh.se (D.G.); goran.stahle@sh.se (G.S.); 4Department of Health Sciences, Swedish Red Cross University College, Hälsovägen 11, 141 57 Huddinge, Sweden; lars@rkh.se (S.L.); sabf@rkh.se (F.S.); 5Division of Insurance medicine, Department of Clinical Neuroscience, Karolinska Institutet, Berzelius Väg 3, 171 77 Stockholm, Sweden

**Keywords:** social participation, dimension, mental health, refugees and asylum seekers

## Abstract

Social participation plays a key role in the integration of refugees and asylum seekers into their host societies, and is also closely tied to the mental health of those populations. The aim of this scoping review was to study how the concept of social participation is described in empirical research, and how it is associated with mental health outcomes. Methods: In total, 64 studies were identified through searches in PubMed, PsycInfo, and Sociological Abstracts. These studies describe various forms of social participation among refugees and asylum seekers, and 33 of them also addressed various forms of mental health outcomes. Results: The identified studies described forms and conditions of social participation—both in the host country and transnationally—that could be synthesized into three broad dimensions: (1) Regulatory frameworks, conditions and initiatives; (2) Established societal organizations and social structures; and (3) Community organized groups. Each of these consisted of several sub-domains. The identified dimensions of social participation were also associated with psychosocial well-being and decreased psychological distress. Conclusions: There is a need for policies to enable and support the participation of refugees and asylum seekers in various dimensions of social structures in host societies. Social participation enhances resilience, re-establishes social lives, and acts as a protective factor against poor mental health outcomes.

## 1. Introduction

The global number of refugees fleeing political instabilities, war, conflict and persecutions has increased substantially in the past two decades from about 15 million in 1995 to 23 million at the end of 2016, an increase that includes 5.3 million Palestinian refugees under the United Nations Relief and Works Agency mandate. There was an unprecedented increase in the number of asylum seekers from about 500 thousand to 2.5 million during the same period. This global influx of refugees and asylum seekers (hereafter referred to as RAS) makes the humanitarian and resettlement response challenging for the host countries [1] and actualizes the need for a knowledge-base to inform policy and practice in these regards.

The high influx of RAS across the world has highlighted the following two important issues. First, the social integration of refugee populations has emerged as a primary political agenda as well as an issue of controversy in the socio-political discourses of many host countries [2]. Second, given the extreme adversities often characterizing the course of forced migration and their associations to adverse mental health outcomes [3], specific barriers to health care experienced by this group are of particular concern [4,5,6]. Also, the potential reciprocal influences between mental health and post-resettlement social conditions are being increasingly acknowledged [7], which suggests the necessity of an integrated approach to issues of mental health and social integration among refugees.

### 1.1. Mental Health of RAS

The higher rates of mental health problems among RAS compared to other groups of migrants have often been ascribed to episodic or long-term exposure to extreme life adversities caused by pre- and peri-flight events [6,8,9,10]. In addition to the prevalent focus on the detrimental effects of previously experienced traumas [6], a more recent focus addresses the impact of post-migration contextual and social factors as important determinants of mental health among refugee populations [7,11]. Post-migration contextual factors include, among others, the loss of social networks, unemployment, poverty and a lack of access to basic resources, perceived discrimination, increased family violence, and difficulties navigating settings of resettlement [6,11,12]. This host of profoundly challenging factors constitutes an important matter of concern for both migration policy and mental health research and work, as disadvantaged post-migration conditions contribute to the deterioration of mental health and generate new significant life stressors [10]. 

The legal status of refugee has a humanitarian grounding with the primary purpose of providing immediate shelter [13]. Perhaps as a consequence of this, policies and discourses around long-term refugee integration rarely encompass psycho-social needs for stability and security or identity [14,15]. Host countries’ institutional structures, legal systems, as well as a current increase in hostile sociopolitical discourse impact the preconditions for achieving these long-term needs by modulating accessibility and availability of, i.e., work, education, housing and civic participation [16]. Moreover, initial vulnerability to mental health issues, such as PTSD, associated with forced migration may have adverse impacts on refugees’ social network orientation [17], and as such increase the risk for social isolation in the new host society. 

In general, a large body of literature highlights active participation in social contexts as a factor that bears substantive potential to promote health—particularly mental health, well-being and positive health-related behaviors [18,19,20,21,22]. There is, furthermore, an increasing global call for adopting social inclusion or integration as key outcomes for those with mental disorders [23]. Social participation, a key concept denoting participation in social events, structures and organizations, has been suggested to be strongly linked to the formation of social capital [24], and thus carry health beneficial effects. Parity of participation, that is, social arrangements that enable all members of a society regardless of status to equally participate in all spheres of social interactions and social life, also has been viewed to be at the core of social justice [25].

The concept of social participation in regard to refugees has, furthermore, been put forward as a conceptual alternative to the politically charged, and therefore unclear, concept of social integration [16]. Currently, a number of concepts such as assimilation, acculturation, adaptation, incorporation, inclusion/exclusion, insertion, settlement, denizenship, citizenship, and, particularly, integration have been used in both research and practice to refer to the process and/or the outcome of refugees’ relations to the new social environments of the host society [14,16]. Discerning the active role of refugees on the basis of these process- or status-oriented constructs, however, may prove to be difficult as many of the concepts imply multiple-way interactions and involve a host of stakeholders [16]. From this point of view, *participation* as a framework refers to both a *structure*, i.e., access of the refugees to social domains; as well as *agency* through ascription of an active role to refugees [26].

### 1.2. Gaps in Knowledge and Rationale for This Study

Despite the advantages that the framework of *social participation* may provide, the concept still lacks a unified definition in the context of refugees’ post-resettlement social conditions. The insufficient conceptual clarity also impacts the issues of assessment and measurement. Although assessment of refugee participation within specific social domains, such as labour market and education, may be more readily accessible by aggregated data such as Migrant Integration Policy Index, or community employment rates [27], the broader modes of refugees’ active involvement in multiple and inter-related levels of society and social structures remain elusive. 

It is therefore important to review the extant empirical research on refugee social participation, in order to identify overarching core structures, areas and contexts that may be used to delineate a model of the concept. Such a review may also provide opportunities for methodological development by presenting adequately quantifiable facets of social participation. 

In this review, we address how social participation is described, used, and/or operationalized in empirical research with particular focus on RAS. In addition, we explore the pathways by which different facets of social participation are described as associated with mental health outcomes among RAS.

## 2. Materials and Methods

Based on a scoping review methodology [28,29], the present review explored how the concept of social participation is described and utilized in extant empirical research among the RAS populations. The choice of methodology was primarily motivated by the disparate and non-systematic use of targeted concept of *social participation,* and the resulting scarcity of robust studies on associations between the construct and mental health among the population of interest. Therefore, the focus was directed toward identifying empirical research that assesses a *social structure* (i.e., community, school, etc.) and an overt form of activity (i.e., *involvement, engagement* etc.), as well as the ways in which mental health has been addressed in such studies. Thus, the searches were chosen to encompass combined terms related to social participation, such as *community engagement* and *civic participation.* Other theoretical concepts that denote proprieties or function of social structures without including an explicit form of activity (such as social support and social capital) were not included in the search. 

### 2.1. Search Strategies

Three bibliographic databases were searched: Sociological Abstracts, PsychInfo and PubMed. The literature search was conducted in all three databases on 19 September, 2017. The search results were then transferred to the reference management software, EndNote. An initial selection of relevant publications was made by screening of titles and abstracts. Each abstract was screened independently by two researchers and discussed thereafter until consensus was reached. 

The inclusion criteria for abstracts were as follows: (1) the study has been published in a peer-reviewed scientific journal; (2) the study population is RAS; (3) the study is on the topic of social participation or another one of the categories/concepts named above—among RAS. Relatively broad inclusion criteria were used at this stage in order not to miss relevant publications. Another stage of selection was conducted by reading the full text of those publications that had been included via abstract screening. Again, each full text was screened independently by two researchers, and disagreements were discussed until consensus was reached.

The following exclusion criteria were applied: the study includes migrants, immigrants, Internally Displaced Persons (IDP), not specifying (explicitly mentioning) RAS; the publication format is dissertations, books, conference papers, commentaries, editorials, letters; the paper does not deal with the social engagement of refugees; the article addresses passive membership in a community defined solely by attributes such as place, territory or nationality, residence, etc., and does not implicate common activities; the article describes episodic social events such as weddings, family gatherings, birthday parties, etc. Language: not in English.

### 2.2. Data Extraction and Quality Assessment

The articles that were included after full-text screening were summarized through a systematic data extraction. A data extraction form was constructed that could encompass both quantitative and qualitative studies, as well as studies from various academic disciplines. This stage was also conducted independently by two researchers, and data were extracted regarding the following: study methods, background variables among the study population, data regarding how social participation and mental health were defined and addressed in the studies, and information about the specific domain of social participation that was addressed. The “domain” in this regard refers to the conditions for, or the specific activities addressed in relation to social participation. 

Based on the extracted data, the studies were summarized in relation to the study purpose, according to the following: (a)Concepts and methods addressed regarding social participation among RAS.(b)Associations between social participation and mental health among RAS.

We conducted a quality assessment of the included studies: The included quantitative studies consisted mainly of cross-sectional studies. Therefore, the assessment tool “Quality Assessment Tool for Observational Cohort and Cross-Sectional Studies” was used [30]. The qualitative studies included were assessed using six key questions proposed by Kuper et al. [31]. The results of the quality assessment are shown in Appendix A.

## 3. Results

The search process for relevant studies included in the present review is presented in Figure 1. The found references (n = 146) were exported from EndNote to Covidence—a software designed to improve the production and use of systematic reviews for health and well-being. In total, 59 of the screened articles were found via Sociological Abstracts, 48 were found via Psychinfo and 39 via PubMed and 9 duplicates were excluded. The remaining 137 studies were selected for abstract screening. Thirty-five studies were excluded based on the selection criteria for abstract screening and 102 studies were assessed through full text, of which 38 were excluded. Finally, 64 studies were included for data extraction. 

### 3.1. Description of the Studies

A description of the 64 studies selected for analysis is shown in Table 1. Fifty studies are built on qualitative methods, of which 37 were based on interviews involving between 5 and 150 participants. In seven studies, the interviews were conducted in combination with ethnographic fieldwork as well as document and literature analysis. The interview forms were most often semi-structured or focus groups. Only a few studies were based on unstructured interviews. In addition, there were some case studies that did not specify the nature of the empirical material. A total of eight studies were based on quantitative data only, with a participant number varying from 150 to 1600. A total of five studies combined quantitative and qualitative methods. Quantitative data were, in some cases, collected by the researchers themselves, but data from national population surveys and health records were also used. The majority of the quantitative studies were based on cross-sectional survey design. Most of the studies were conducted among RAS populations residing in urban areas. Studies dealing with Palestinian RAS were conducted in refugee camps. 

### 3.2. Social Participation among RAS—Definitions and Conceptual Model

The empirical articles included in this overview addressed a relatively large number of theoretical constructs with a varying degree of definition specificity. However, by including both a social structure and an activity, these constructs were classified under the umbrella concept of social participation. The included constructs referred to social structures that encompassed social conditions and frameworks, institutions and organizations, as well as informal communities. A conceptual model of social participation on the basis of the included structures in which the active participation was described was then outlined. The model consisted of three main *dimensions* each denoting multiple social *domains* for RAS’ active participation. The three main dimensions are labelled:Regulations and frameworks, conditions and initiativesEstablished societal organizationsCommunity organized groups

As shown in Figure 2, Dimension 1 (Regulatory Frameworks, Conditions and Initiatives) is described as the foundation of and determining conditions of the other two dimensions of social participation. Dimension 2 (Established Societal Organizations) includes domains of participation that are initiated by host society’s authorities or established institutions. Dimension 3 (Community Organized Groups) includes domains of participation that are initiated by RAS themselves.

The bi-directional arrows between Dimension 2 and 3 indicate potential overlaps between the *Established Societal Organizations* and *Community Organized Groups* dimensions. The overall conceptual model, thus, proposes a dynamic interaction between the three main dimensions of social participation rather than an exhaustive and exclusive checklist of social frameworks for RAS participation.

### 3.3. Dimension 1: Regulatory Frameworks, Conditions and Initiatives

This dimension includes studies that address the host country’s laws, policies and organized initiatives in relation to RAS. Foremost, the legal status of refugees in the new country, as well as the state-coordinated initiatives for reception and integration, are important domains of this dimension. Examples of such initiatives include state-initiated language education and labor market introduction.

#### 3.3.1. Domain 1.1. Legal Status, Conditions and Incentives

Refugees face opportunities, barriers, and facilitators for participation in the key social spheres in the host country, that are primarily dependent on the overall legal status and laws that regulate refugees’ access to the host country’s social structures. Examples of this include receiving residence permit and citizenship, which provides the rights to participate in educational programs, employment and health care [32,33,34,35,36,37]. Having a permanent residence permit, and especially citizenship, furthermore increases the individual’s sense of belonging to the new society and facilitates active participation in the main social spheres of the new country [32,33,34,35]. On the other hand, prolonged asylum processes and temporary residence permits hinder the development of a sense of belonging, and represent important barriers to accessing important societal institutions [35,36,37].

#### 3.3.2. Domain 1.2. Reception Systems and Targeted Actions

Well-functioning policies in regard to refugees’ resettlement, including reception systems for asylum seekers, are also important prerequisites for active participation in the host society [33,34,37,38,39,40]. In fact, reception programs allow access to societal systems, including language courses [41], which in the long run create opportunities for entry into the labor market [42]. However, reception programs depend on the success of government initiatives concerning, for example, providing housing and creating opportunities for making refugees employable [34].

#### 3.3.3. Domain 1.3. Conditions for Political Engagement

National migration and integration policies form the fundamental basis and conditions that limit or facilitate refugees’ possibilities to actively take part in key domains of society. However, refugees lack formal opportunities for influencing the very regulations that determine their social conditions and future in the host country [43,44]. Nevertheless, engagement in established civil society is often the sole opportunity to achieve and exercise some degree of such influence [43,44]. According to two studies, however, civil society initiatives and organization may often risk underutilizing the possibilities to mobilize refugees’ own capacity to assert an impact on the local policies [43,44].

### 3.4. Dimension 2: Established Societal Organizations and Social Structures

This dimension included studies that address RAS’ active participation in host society’s social structures which can solely be accessed by adopting explicit public and formal roles (i.e., as employed, patient, student). These host society’s structures, and the roles required and upheld within them, are defined by established societal organizations that generally function unconditionally of RAS’ specific circumstances. In regard to such social structures, participation has mainly been studied within the domains of the labor market, financial transactions, education and health care. Therefore, the focus here lies on those activities that are organized for RAS within the public sector.

#### 3.4.1. Domain 2.1 Labor Market

Labor market participation is an important aspect for refugees in a new country [38,45,46,47,48,49]. Unemployment as a main indicator of labor market exclusion, however, cannot solely be explained by intentional poor societal engagement on the part of refugees. For example, in times of economic recession in the host country, refugees who have learned the host country language (an indicator of societal engagement) are nevertheless not ensured improved access to the labor market [49]. Labor market inclusion is also determined by a number of other factors such as legal status [36], asylum seeking processes [35,37,42], household structure [50,51], traumatic experiences and consequent poor mental health status [35,37,51,52] and the non-validation of education or work experience refugees have obtained in the country of origin [38].

#### 3.4.2. Domain 2.2 Financial Transactions

Participation in this domain concerns the utilization of, as well as interactions with, financial systems at a group level (i.e., refugees in camps) or individually. Although it is debated whether economic progress of refugees is a pertinent indicator of the extent of participation in the host society [53], some authors view economic sustenance as a basic condition for participation and tied to citizenship issues [32,35,54]. Thus, taking part in financial transactions affects the possibilities for a meaningful everyday life and refugees’ health [36,37]. Interestingly, the studies that dealt most explicitly with financial transactions were from the late 1980s and explored the relation between income and psychological well-being [55], as well as economic adaptation, measured, for example, by ownership of household appliances and strategies to afford them, such as saving money or taking loans [49]. Two studies were focused on the effects of public financing. One concerned the effects on refugees networking within or outside of ethnic groups, depending on the size of allowances to refugees from the state [34]. The other undertook a discussion of how an economic program (conditioned by health and educational factors) among refugees in a settlement, intended to spur the participation of refugee women outside of the household. The program did not have the intended effects because rather than counteracting gender hierarchies the larger sums given to women and girls tended to foster community division [56].

#### 3.4.3. Domain 2.3 Education

Refugee participation in different educational programs offers a possibility for building social networks [57,58,59,60,61]. Education is a prerequisite for participation in central social spheres of the host society, but also constitutes an important domain of active participation in itself [33,36,41,59,62]. Furthermore, acquiring host country language proficiency has been suggested to be an important source of self-esteem [62], as well as allowing communication across ethnic groups and across generations [58]. On the other hand, a focus on language courses has also been critiqued to be used as a tool for serving an idea of a “uniform population”, tacitly based on nationalist ideas of a shared common language within the nation [35]. One study also shows that a one-sided focus on proficiency in the host country’s language can undermine the importance of other qualifications [38]. Along the same lines, while speaking merely one’s mother tongue may constitute an obstacle to participation in some social domains [33,63], other studies point out the benefits of a shared language within the own group [55,64]. Participation in education may also reflect other facets of social conditions, as refugees may at times experience barriers to participation in education due to family and household responsibilities [65] as well as fear of discrimination and social dismissal from the majority population within the education sector [63,66].

#### 3.4.4. Domain 2.4 Healthcare

Health care access and good health are prerequisites for active participation in key social spheres of the host society [67]. Beyond this, health care utilization and contacts with health care delivery systems constitute, in and of themselves, also a form of involvement and participation with an important social structure [68]. Active participation in other domains, such as work and educations, may further facilitate health care access and utilization. However, RAS experience a number of barriers in accessing health care systems. For example, communication between health care service providers and patients is of key importance for building trust in health care institutions, where insufficient health literacy and language issues may specifically constitute important barriers in this regard [69]. Additional refugee-related factors such as socio-economic position in the host society, gender roles, lack of adequate information, difficulties in navigating organization of the health care services, and cultural beliefs and practices in regard to health and illness may potentially hamper access to health care utilization [57,65,69,70,71,72,73]. Engagement of the refugee communities in the development of health care policy and delivery, as well as participatory health care research, are suggested to be viable for addressing such barriers to care [69,71,72,74,75].

### 3.5. Dimension 3: Community Organized Groups

This dimension concerns participation in social structures which the participant enters in an informal role, such as refugees’ personal networks and social resources. These structures, and the roles upheld within them, are defined by refugees’ own communities and to a higher degree shaped by refugees themselves and based on subjective perceptions of belonging and connection with others. The dimension includes participation in non-governmental institutions, such as voluntary organizations and associations, sports, or cultural and religious activities.

#### 3.5.1. Domain 3.1 Ethnic Communities

Participation in communities that are defined by ethnic or cultural background includes participation in voluntary associations as well as less formalized activities based on shared ethnicity. This domain of participation is associated with, for example, increased political engagement in the host country as well as in the country of origin. A characteristic of this domain that is often addressed in the studies is the improved well-being associated with the sense of belonging to a community [53,63,64,77,78]. In addition, participation in trans-national ethnic networks can have impact on an international level in the form of, e.g., remittance as well as political engagement in the issues relating to the country of origin [53,79,80]. Furthermore, participation in this domain is suggested to reinforce cultural identity and provide access to networks that can lead to employment, housing, and other social benefits by means of using resources within the ethnic communities [53,55,78,81].

#### 3.5.2. Domain 3.2 Religious Congregations

Activities defined by religious affiliation are also described as important domains for participation. Several included studies describe churches, mosques and temples as places where individuals may receive opportunities to reconnect to practices and resources that are specific for the community, for example rituals and celebration of holidays. In some instances, the religious group may gather and meet on a regular basis, and provide a platform for interaction and practical help for members of the (sometimes ethnically organized) congregations [51,62,82,85]. As a consequence, participation in religious activities can be a source of emotional support, meaning-making and acceptance. Several included studies also show that practical aids, such as providing food and transportation to church are decisive factors, in the sense that participation in settings other than school, for example, is facilitated when religious organizations provide such forms of support [62].

#### 3.5.3. Domain 3.3 Sports and Leisure Groups

Sports and leisure are identified as among the most important social activities for refugees outside of school and work [62,84]. Participation in such activities may lead to beneficial effects, through promoting cohesion among ethnic groups [53], a sense of belonging and identity [59], as well as providing a basis for conflict resolution [60,85]. However, socio-economic factors and lack of family support are important barriers to participation in this domain, especially among female refugees [62,86].

#### 3.5.4. Domain 3.4 Political Associations

Political engagement as participation can take place both at a national and transnational level. At the host country national level, participation and party membership is often connected with an improved sense of belonging and well-being [87], although political engagement among RAS is often mainly focused on migration issues and/or politics in the country of origin [88]. Barriers to political engagement in the host country include a lack of sense of belonging, as well as socio-economic exclusion [85,87]. In two of the articles, political activism beyond national borders is discussed from the perspective of the sense of responsibility that refugees feel toward the home country, and the incentives and hindrances formed by host country institutions in relation to these engagements [88,89]. The complexity of transnational political engagement is emphasized in two studies, as the kind of activism that can complicate intergovernmental relations, when the host country’s foreign policy is used as a channel to pressure homeland governments [44,79]. Such engagements in the politics of the home country can also be understood as civic participation in the host country’s (foreign) politics [44].

#### 3.5.5. Mental Health Classifications

This review identified 33 studies that addressed the associations between a broad spectrum of various forms of mental health and well-being-related concepts and social participation among RAS. A summary of key findings from each study is reported in Table 1. The included concepts were categorized on the basis of the common contents in four mental health domains labeled as follows: (1) psychosocial well-being, (2) psychological well-being, (3) psychological distress and (4) psychiatric disorders. A description of each category, as well as the associations between the dimensions of social participation with these categories are presented below.

1. **Psychosocial Well-being**

This category includes described aspects of well-being that are directly related to RAS’ subjective experiences and appraisals of social ties and relational attachments. As such, psychosocial well-being bears an evident association with social participation. This category, thus, describes well-being as related to attachments to places and people, affective relationships, a sense of belonging in a narrower social context, as well as a sense of belonging in the receiving host society at large.

Attachment to people and places in both RAS’ origin and the receiving country was suggested to be mainly related to social and emotional ties to relatives and/or friends from the same country of origin [41,62]. Relationships with those from same origin in resettlement contributes to such well-being by allowing for exchange of common experiences [55,57,63,90]. Another important aspect contributing to psychosocial well-being or lack of such was the sense of belonging in the host country in a broader sense. Sense of belonging was, moreover, associated with engagement in civic rights activism [90], educational programs [91] and participation in other formal and informal social domains [34,38,48,58,82]. On the other hand, refugee status in contrast to other immigrants [32] and long process of asylum applications were found to have a negative effect on psychosocial well-being [37] through deprivation of sense of attachment and belonging.

2. **Psychological Well-Being**

The concept of well-being as described in this category implies the positive dimension of mental health that is more pronouncedly related to subjective personal well-being, perceived self-efficacy, and a strong sense of autonomy. This category could be further divided in the subcategories of (1) *general well-being and personal competency*; and (2) *existential well-being*.

Important aspects relating to psychological well-being included a sense of personal competency, a sense of identity [90], self-esteem, empowerment [58], and resilience [59,91]. A sense of identity was found to be strengthened by engagements in, for example, civic rights activism [90]. Self-esteem, empowerment [58] and resilience were enhanced by participation in different school programs [59,91]. Also, psychological well-being was found to be enriched by community support programs [55] and recognition of one’s human capital [48]. Participation in social activities such as religious activities was associated with a sense of acceptance of one’s life situation [82], and community gardening was associated with increased social support and a sense of hope [92].

3. **Psychological distress**

This category delineates various forms of subjective distressful emotional states, such as stress, shame, sadness, and anxiety. Although psychological distress may be related to and overlap with psychiatric disorders and psychopathological states, an important conceptual difference between the two concepts has been evident in the included studies in this review. Accordingly, psychological distress here denotes non-pathological emotional responses to adversities experienced by RAS.

Psychological distress and anxiety were associated with uncertain residential status and long processing times of applications for asylum [34,35,37]. The lack of either government or volunteer programs promoting different forms of social activities [34,40,62], and devaluation of RAS’ human capital, i.e., education and professional skills [38] contribute to feelings of frustration and anxiety. Furthermore, lack of opportunities for social engagement or experience of social exclusion were associated with psychological distress [67,93]. On the other hand, community gatherings [57] and involvement in leisure activities [94] were reported to function as protective factors against depressive states and experiences of anxiety. In relation to healthcare, community-based approaches in psychiatry help to address sociocultural barriers to psychiatric care [72] and the quality of health care service delivery [52], resulting in increased access to preventive healthcare and consequent decreases in psychological distress [68,73]. Additionally, a community-based approach in research enables participants to address sensitive issues related to health risk behaviors [84] and emotional experiences, and this in turn results in decreased stress [73] and depression [75].

4. **Psychiatric disorders**

This category concerns diagnostic entities that are formally identified and classified as psychiatric or mental disorders. Because of reliance on standardized classifications, the few studies that include this category do not provide specific definitions or operationalizations of the included mental health concepts.

According to an epidemiological study of first admission to psychiatric hospitals, differences in the rates of schizophrenia, paranoia and other psychiatric disorders between groups of refugees, were found to be partially explained by differences in degrees of social isolation, which is associated with a sense of marginalization and decreased access to social support [84]. Another study assessing psychiatric disorders among refugees’ parents who had experienced the death of their children, did not show increased rates of psychiatric disorders despite less engagement in social activities [51].

## 4. Discussion

The aim of this study was to systematically review the concept of social participation among RAS as it is described, utilized, and assessed in empirical research dealing with RAS. A further aim was to explore the associations between social participation and mental health of RAS in the reviewed literature. This review identified a total of 64 relevant studies that described various forms of social participation among RAS, of which 33 studies also addressed mental health. The included studies were from a spectrum of disciplinary fields, ranging from political science and sociology, to public health and psychology; and concerned a plethora of different refugee populations in various settings ranging from refugee camps to resettlement contexts such as schools and ethnic communities. Despite this broad diversity, common themes in the included empirical studies could be discerned, on the basis of which a conceptual model of social participation of RAS could be constructed.

In broad terms, the conceptual model presented in the results could be used to define the concept of RAS’ social participation as: *access to, and active involvement in key social dimensions within the host societies, including*
*not only*
*the host society’s*
*formal and established organizations and institutions, but also in social settings constructed on the basis of refugee communities’ own shared characteristics, preferences, and needs.* The latter, which includes social settings based on shared ethnic backgrounds, religion, political orientations, and even leisure activities, constitutes an important dimension of social participation as it potentially provides important instrumental and emotional resources that are embedded in, and accessed through, participating in social networks [19]. This dimension, however, may risk being neglected or overshadowed by an over-emphasis on the concept of social integration [16,95], which mainly builds on participation in receiving societies’ formal institutions such as labour market and education. Whereas, according to the model in this review, accessing and entering into established host societies’ social contexts remains a core dimension of RAS’ social participation, our results suggest that a comprehensive approach to social participation needs to incorporate social domains and dimensions that stem from RAS’ own conditions, needs, and shared characteristics.

By including the concept of *access* into the empirically based model and the extracted definition based on the current review, a crucial dimension of social participation is furthermore highlighted that reaches beyond individuals’ and communities’ social actions. Our results indicate that the formal legalizations and policies that determine and regulate the formal status of RAS, largely condition the forms and extent of social participation. Although, as opposed to other forms of migration, it is international law that determines the rights granted to refugees [13], national policies regarding asylum seeking procedures and resettlement vary considerably across different receiving countries [96]. Given that access to important social spheres such as labour market, health care, and education, as well as opportunities to create, access, and participate in refugees’ own social settings are strongly modulated by such policies and regulations, this dimension could be theoretically viewed as a form of *structure* [97,98] depending on which the social action of RAS and their communities could be considered as expressions of *agency* [97,98]. Although the dynamics and interrelations between structure and agency remain theoretically debated [26], RAS’ extremely sparse formal opportunities to bring about substantive change in such formal legislations and policies suggest that a model of social participation for forced migrant populations needs to include this dimension as a fundamental defining feature. In practice, this also highlights the fallacies of extreme individualistic and current populistic social discourses that ascribe the full extent of social participation to individual choices and actions [99]. In contrast, parity of participation in regard to RAS implies a need for policies that ensure that the forced migrant populations’ perspectives are represented, recognized, and valued in all relevant social structures [100].

In this review, we also examined the studies that describe different ways in which various dimensions of social participation were studied as associated with mental health. The included studies presented a broad spectrum of phenomena that were deemed to reflect different facets of mental health among RAS. In general, these forms or facets of mental health fell into two broad categories: either as descriptions of positive states of well-being, or delineated as psychological distress as well as, more rarely, psychiatric disorders. A general trend among the studies was implicit or explicit narratives of a causal link between social participation and improved subjective and psychosocial well-being as well as decreased psychological distress. However, mental health was not only described as the result of social participation and as influenced by regulations and frameworks that condition such participation, but also to some extent as a factor that influences opportunities for inclusion in various social contexts [34,38,48,58,82,90,91]. These findings are in line with previous studies in both refugee- and non-refugee populations that suggest that, aside from being an indicator of integration [16] social participation also promotes health, well-being and positive health-related behaviours [18,19,20,21]. A possible pathway of such a health promotive effect may be through access to different forms of everyday life social resources that help RAS re-establish social lives in their new context [19,57,59]. Other protective factors stemmed from, or related to, social participation in this respect include resilience gained through an increased sense of belonging, self-esteem and empowerment, which are all linked to social participation [58,59,90,91].

Furthermore, the scarcity of studies that include psychiatric disorders [51,84], either as an antecedent or as sequalae of poor social participation, indicate that empirical research into the social conditions of RAS on one hand, and clinical forms of mental health issues on the other hand, remain in disjunction. Apart from recent epidemiological studies that have assessed the risk for labour market marginalization associated with mental disorders among refugees [101], the lack of studies may be a result of the lack of a comprehensive and measurable conceptualization of social participation. Given the requirement of valid assessment and effective markers of both outcomes and explanatory variables in clinical and epidemiological studies [102], an empirically based model and definition of social participation, as presented in the current review, may serve as a basis for development of assessment methods for social participation.

### Limitations and Strengths

A limitation of the present review is that within the mapped research field, the concepts related to social participation among RAS lack references to clear and specific definitions, resulting in some ambiguity in regard to the targeted construct. However, this lack of unified definition is also the primary motivation for conducting the review.

The review also shows that many of the studies in this field suffer from methodological weaknesses which inevitably translates to a limitation in regard to the results of the review. This can be partly explained by the practical difficulties in recruiting participants, conducting studies on a larger scale with repeated data collections and representative samples of refugee populations. The lack of robust and rigorous methodological approaches and clear definitions and measurable indicators of social participation are research areas, where the present review identifies important needs for further research.

The present review also has strengths: The review is based on comprehensive multidisciplinary sources without limitations in time period or geographical location and includes both qualitative and quantitative studies. Based on these extensive sources that have been systematically reviewed, and the conducted quality assessment of the studies, this review represents a particularly comprehensive approach to an important but neglected construct. Another strength of the review is that despite the lack of clear definition and a wide variety of methods and theoretical and empirical approaches in the material, the review has been able to identify a number of common dimensions and domains that constitute the components of the provided model, which may be used to advance several relevant fields of refugee studies.

## 5. Conclusions

In all, the present review identifies three core dimensions of social participation among refugees, delineates several domains included in each of these dimensions, and provides an empirically based model and definition of the target construct. Furthermore, the review also shows that the concept has been specifically addressed as associated with facets of mental health among RAS, suggesting a need for more systematical examination of the causal directions and potential reciprocity between mental health and social participation among refugee populations. Furthermore, the results of the review indicate a need for elaboration of methodologies required to assess the comprehensive and multi-dimensional construct of social participation.

## Figures and Tables

**Figure 1 ijerph-16-04027-f001:**
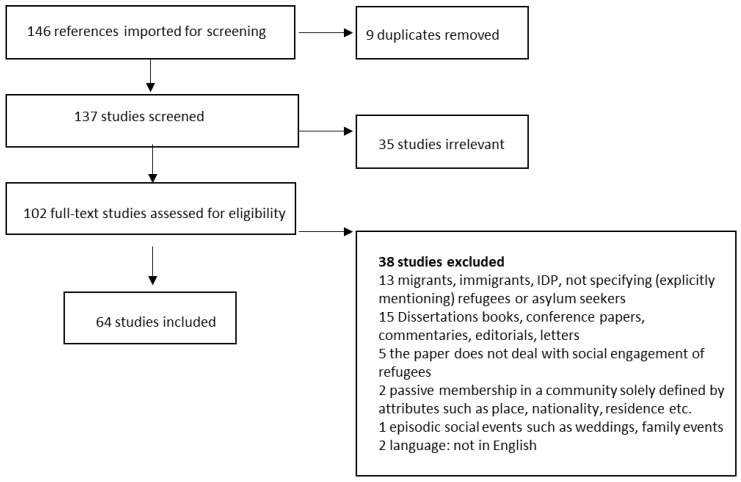
Flow-chart of the search process.

**Figure 2 ijerph-16-04027-f002:**
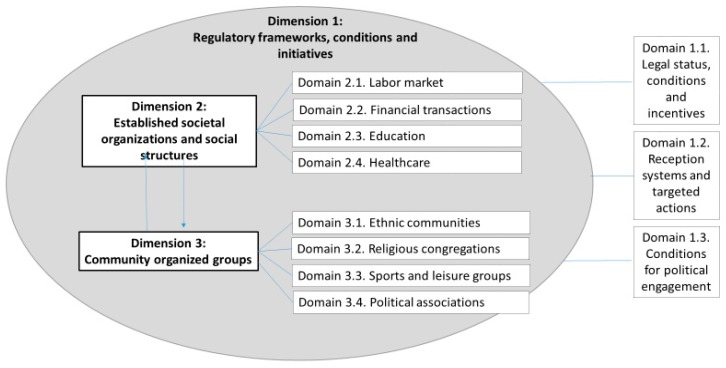
Dimensions and domains of social participation among RAS.

**Table 1 ijerph-16-04027-t001:** Descriptive table of included studies.

First Author, Year	Country of Origin and Host Country (n)	Social Participation Domain	Mental Health Domain	Method Used for Assessment
Khawaja, 2006 [19]	Origin: Palestinians Host: Jordan (n) 1615	Community organized groups: Ethnic communities	Psychological distress and Psychological well-being	Survey
Brown, 2011 [32]	Origin: Liberia Host: USA (n) 25	Regulatory frameworks, conditions and initiatives: Legal status AND Established societal organizations: Financial transactions	Psychosocial well-being	Participant observation
Da Lomba, 2010 [33]	Origin: not spec. Host: Great Britain (n) not spec.	Regulatory frameworks, conditions and initiatives: Legal status and Reception systems AND Established societal organizations: Education	-	Document analysis
Korac, 2003 [34]	Origin: Former Yugoslavia Host: Italy; Netherlands (comparison) (n) 60	Regulatory frameworks, conditions and initiatives: Legal status and Reception systems AND Established societal organizations: Financial transactions	Psychosocial well-being and Psychological distress	Interviews
Stewart, 2014 [35]	Origin: various nationalities, 23 in total Host: Scotland (n) 30	Regulatory frameworks, conditions and initiatives: Legal status AND Established societal organizations: Financial transactions and Labor market and Education	Psychological distress	Interviews.
Baban, 2016 [36]	Origin: Syria Host: Turkey (n) 30	Regulatory frameworks, conditions and initiatives: Legal status AND Established societal organizations: Labor market and Financial transactions and Education and Healthcare	-	Interviews and participant observation
Ingvarsson, 2016 [37]	Origin: Afghanistan, Iran and Iraq Host: Iceland (n) 9	Regulatory frameworks, conditions and initiatives: Legal status and Reception systems AND Established societal organizations: Labor market; Financial transactions AND Community organized groups: Sports and leisure	Psychosocial well-being and Psychological distress	Semi-structured interviews
Ricento, 2013 [38]	Origin: Colombia Host: Canada (n) 1	Regulatory frameworks, conditions and initiatives: Reception systems AND Established societal organizations: Labor market and Education	Psychological distress and Psychosocial well-being	Interview, case study
Stewart, 2015 [39]	Origin: Sudan and Zimbabwe Host: Canada (n) 85	Regulatory frameworks, conditions and initiatives: Reception systems	Psychological distress	In-depth interviews
Hynes, 2009 [40]	Origin: not spec. (15 countries) Host: Great Britain (n) 48	Regulatory frameworks, conditions and initiatives: Reception systems	Psychological distress	Interviews and focus group discussions
Barnes, 2001 [41]	Origin: Vietnam Host: Australia (n) 14	Regulatory frameworks, conditions and initiatives: Legal status and Reception systems AND Established societal organizations: Education	Psychosocial well-being	Semi-structured interviews
Hagelund, 2009 [42]	Origin: not spec. Host: Norway (n) 344	Regulatory frameworks, conditions and initiatives: Reception systems AND Established societal organizations: Labor market	-	Survey and in-depth interviews
Shindo, 2009 [43]	Origin: Kurdish, country not spec. Host: Japan (n) not spec.	Regulatory frameworks, conditions and initiatives: Individual impact	-	Descriptive case study
Horst, 2013 [44]	Origin: African horn (Kenya) Host: Europe (Norway) (n) not spec.	Regulatory frameworks, conditions and initiatives: Conditions for political engagement AND Community organized groups: Political associations	-	Semi-structured interviews
Neudorf, 2016 [45]	Origin: not spec. Host: Canada (n) 12	Established societal organizations: Labor market	-	Semi-structured interviews
Valtonen, 1998 [46]	Origin: Iran, Iraq and Kuwait Host: Finland (n) 29	Established societal organizations: Labor market	-	Semi-structured interviews
Valtonen, 1999 [47]	Origin: Vietnam Host: Finland and Canada (n) 45	Regulatory frameworks, conditions and initiatives: Reception systems AND Established societal organizations: Labor market AND Community organized groups: Ethnic communities and Religious congregations	-	Semi-structured interviews and participant observation
Fozdar, 2011 [48]	Origin: Former Yugoslavia (mainly Bosnians), Middle East (mainly Iraqis) and Africa (mainly Somalis and Ethiopians) Host: Australia (n) 150	Established societal organizations: Labor market	Psychosocial well-being and Psychological well-being	Semi-structured interviews and survey with structured interviews
Montgomery, 1987 [49]	Origin: Vietnamese Host: Canada (n) 537	Established societal organizations: Labor market and Financial transactions	-	Survey
Bach, 1986 [50]	Origin: South-East Asia Host: USA (n) 1500	Established societal organizations: Labor market	-	Structured telephone interviews
Caspi, 1998 [51]	Origin: Cambodja Host: USA (n) 161	Established societal organizations: Labor market AND Community organized groups: Religious congregations	Psychological distress and Psychiatric disorders	Interviews and survey
Ramaliu, 2003 [52]	Origin: not spec. Host: Canada (n) not spec.	Established societal organizations: Financial transactions and Labor market and Healthcare	Psychological distress	Document analysis
Abraham, 2004 [53]	Origin Sri Lanka (Tamil) and Caribbean Host: Canada(n) not spec	Established societal organizations: Financial transactions AND Community organized groups: Ethnic communities and Sports and leisure	-	Unstructured interviews, literature
Preston, 1992 [54]	Origin: Melanesia, Iran Jaya/Indonesia—Melanesia Western Papua Host: Papa New Guinea (n) not spec.	Established societal organizations: Healthcare and Financial transactions	-	Interviews and document analysis
Thanh Van, 1987 [55]	Origin: Vietnam Host: USA (n) 160	Established societal organizations: Financial transactions and Education AND Community organized groups: Ethnic communities	Psychosocial well-being and Psychological well-being	Survey
Gil-Garcia Ó, 2016 [56]	Origin: Guatemala Host: Mexico (n) 148	Established societal organizations: Financial transactions	-	Participant observation, structured and semi-structured interviews
Gerber, 2017 [57]	Origin: Nepal (Bhutanese) Host: USA (n) 50	Established societal organizations: Education and Healthcare	Psychological distress and Psychosocial well-being	Semi-structured interviews and quantitative data
Hope, 2011 [58]	Origin: not spec. - broad Host: Great Britain (n) not spec.	Established societal organizations: Education AND Community organized groups: Ethnic communities	Psychosocial well-being and Psychological well-being	Participant observation and interviews
Marsh, 2012 [59]	Origin: Sierra Leone, Ghana, Croatia, Vietnam and Pakistan Host: Australia (n) not spec.	Established societal organizations: Education AND Community organized groups: Sports and leisure	Psychological well-being	Interviews, focus group discussions and field work
Pugh, 2017 [60]	Origin: Colombia Host: Ecuador (n) not spec.	Established societal organizations: Education AND Community organized groups: Religious congregations and Sports and leisure	-	Case study, document analysis
Naidoo, 2009 [61]	Origin: African countries (only spec., Sudan) (n) 37	Established societal organizations: Education	-	Focus group and semi-structured interviews
Earnest, 2015 [62]	Origin: Afghanistan, Democratic Republic of Congo, Ethiopia, Sudan, South Sudan, Iraq, Pakistan and Burma Host: Australia (n) not spec.	Established societal organizations: Education and Healthcare AND Community organized groups: Religious congregations and Sports and leisure	Psychosocial well-being and Psychological distress	Individual and focus group interviews and document analysis
Barnes, 2007 [63]	Origin: Cuba Host: USA (n) 20	Established societal organizations: Education and Healthcare AND Community organized groups: Ethnic communities and Religious congregations	Psychosocial well-being	Semi-structured interviews
Dandy, 2013 [64]	Origin: Sudan, Afghanistan, Philippines, Bhutan, Burma, Vietnam, Iraq, Iran, Egypt, (England?) Host: Australia (n) 54	Regulatory frameworks, conditions and initiatives: Reception systems AND Established societal organizations: Education AND Community organized groups: Ethnic communities andSports and leisure	-	Individual and focus group interviews
Evans, 2011 [65]	Origin: Africa Host: England (n) 37	Established societal organizations: Education and Healthcare	Psychological distress	Semi-structured interviews
Haddad, 2002 [66]	Origin: Palestinians Host: Lebanon (n) 1073	Established societal organizations: Education	-	Survey
Aylesworth, 1983 [67]	Origin: Indochina, Vietnam, Laos or Hmong, Kambodja Host: USA (n) 217	Established societal organizations: Healthcare	Psychological distress	Survey with structured interviews
Yelland, 2014 [68]	Origin: Afghanistan Host: Australia (n) not spec.	Established societal organizations: Healthcare	Psychological distress and Psychosocial well-being	Interviews
Lazar, 2013 [69]	Origin: Somalians (patients, not interviewed) Host: USA, Ohio (n) 14	Established societal organizations: Healthcare	-	Semi-structured interviews
Conviser, 2007 [70]	Origin: Back, Latino, born in Africa Host: USA (n) not spec.	Established societal organizations: Healthcare	-	Interviews and participant observation
Cheng, 2015 [71]	Origin: Afghanistan Host: Australia (n) not spec.	Established societal organizations: Healthcare	-	Document analysis and expert statement
Pejic, 2017 [72]	Origin: Somalia Host: USA (n) 1	Established societal organizations: Healthcare	Psychological distress	Case study
Worabo, 2017 [73]	Origin: Eritrea Host: USA (n) 15	Established societal organizations: Healthcare	Psychological distress and Psychological well-being and Psychosocial well-being	Focus group interviews (secondary data)
Lindgren, 2004 [74]	Origin: Afghans Host: USA, California (n) 5	Established societal organizations: Healthcare	Psychological well-being	Semi-structured interviews
Riggs, 2015 [75]	Origin: Afghanistan Host: Australia (n) 30	Established societal organizations: Healthcare	Psychological distress and Psychosocial well-being	Interviews
Hitch, 1980 [76]	Origin: Polish, Ukraine and Russian (n) 1164	Established societal organizations: Healthcare	Psychiatric disorders	Healthcare data
Al-Ali, 2001 [77]	Origin: Indochina, Vietnam, Laos or Hmong, Kambodja Host: USA (n) 217	Community organized groups: Ethnic communities	-	Participant observation and semi-structured interviews
Bloemraad, 2005 [78]	Origin: Vietnam and Portuguese (Azorean) Host: USA and Canada (n) 147	Established societal organizations: Financial transactions AND Community organized groups: Ethnic communities and Religious congregations	-	Interviews and document analysis
Casier, 2010 [79]	Origin: Kurdish Host: Belgium and Turkey, EU (n) not spec.	Community organized groups: Ethnic communities and Political associations	-	Participant observation and semi-structured interviews
Casier, 2011 [80]	Origin: Turkey (Kurdish) Host: Belgium (n) not spec.	Community organized groups: Ethnic communities and Political associations	-	Semi-structured interviews
Weng, 2016 [81]	Origin: Asia, Africa, Caribbean, majority from Sudan Host: USA (n) 54	Community organized groups: Ethnic communities	-	Semi-structured interviews
Fox, 2012 [82]	Origin: Rwanda Host: USA (n) 14	Community organized groups: Religious congregations	Psychosocial well-being and Psychological well-being	Semi-structured interviews
Hein, 2014 [83]	Origin: Laos (Hmong) Host: USA	Community organized groups: Ethnic communities	-	Interviews
Makhoul, 2009 [84]	Origin: Palestinians Host: Lebanon (n) 1335	Community organized groups: Sports and leisure	Psychological distress and Psychiatric disorders and Psychological well-being	Survey and focus group interviews
Achilli, 2014 [85]	Origin: Palestinian Host: Jordan (n) not spec.	Community organized groups: Sports and leisure and Political associations	-	Participant observation, semi- and unstructured interviews
Rosso, 2016 [86]	Origin: various Host: Australia (n) 152	Community organized groups: Sports and leisure	-	Survey and semi-structured interviews
Jeong, 2016 [87]	Origin: North Korea Host: South Korea (n)151	Community organized groups: Political associations	-	Survey
Østergaard-Nielsen, 2001 [88]	Origin: Turkey(/Kurdistan) Host: Germany and Netherlands (n) not spec.	Community organized groups: Ethnic communities and Political associations	-	Document analysis
Rivetti, 2013 [89]	Origin: Iran Host: Italy and Turkey (n) 52	Community organized groups: Political associations	-	Participant observation and semi-structured interviews
Grossmann, 2014 [90]	Origin: European Judes Host: USA (n) not spec.	Regulatory frameworks, conditions and initiatives	Psychosocial well-being and Psychological well-being	Narrative interviews
Matthews, 2008 [91]	Origin: Africa, Middle East Host: Australia (n) not spec.	Established societal organizations: Education	Psychosocial well-being and Psychological well-being	Semi-structured interviews
Eggert, 2015 [92]	Origin: not spec. Host: USA (n) not spec.	Established societal organizations: Healthcare	Psychological well-being	Survey, interviews and participant observation
Whiteford, 2007 [93]	Origin: Albania Host: Makedonia (n) not spec.	Community organized groups	Psychological distress and Psychological well-being	Refugee camp
Stack, 2009 [94]	Origin: Afghanistan Host: Canada (n) 11	Community organized groups: Sports and leisure	Psychological distress	Semi-structured interviews

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
