# Peer review of "A Scoping Review and Conceptual Model of Social Participation and Mental Health among Refugees and Asylum Seekers"

_ijerph, 2019, doi:10.3390/ijerph16204027_

Round 1

Reviewer 1 Report

This research provides a very important contribution to understanding the structural dimensions of social participation and potential pathway/linkages to mental health. The paper overall is clearly presented and methods are explained well. Please note a minor typo under Domain 2.4 Healthcare line 5- should read further facilitate not facilitates.

The most valuable contribution of this scoping review is the conceptual development and model of key findings/organization of themes as regulatory frameworks, societal organizations and structures and community organized groups.

It might be useful to consider how this scoping review can be used to inform policy and practice related to better integration, the findings suggest many RAS remain socially excluded. Nancy Fraser's work on political parity of participation advocates that this must be included in liberal democracies in order to promote social justice aims. The authors might consider the value added in this work related to not only institutional, international and national immigration policy but political position as having influence on these policies, e.g. neo liberal vs liberal democratic host countries and how this also shapes inclusion and exclusion.

This review is rigorous and reflects findings from other research in the field. This is a significant contribution as the authors note a gap in knowledge related to social participation and mental health of RAS groups. Most of the literature reviewed includes refugee/migrant women and is discussed as gender as a social dimension of integration, social participation. However, gender is an operating structure which also determines not only women's access but other social sex/gender identity groups. Men in particular have been socially excluded from labour, education based on increased services and resources which target women only.

Overall I highly recommend publication and provide minor suggestions related to making a political recommendation for policy and practice among those who provide social integration including healthcare services which remain individualistic and based on neo liberal models of efficiency.

Author Response

Please see our responses to the reviewer comments in bold letters below: 

AUTHORS’ RESPONSE: We wish to thank the reviewer for the truly encouraging comments on our paper.

Please note a minor typo under Domain 2.4 Healthcare line 5- should read further facilitate not facilitates.

We have corrected this typo, as well as number of other minor typos, the corrections of which can be seen in the tracked changes of the resubmitted manuscript.

Overall I highly recommend publication and provide minor suggestions related to making a political recommendation for policy and practice among those who provide social integration including healthcare services which remain individualistic and based on neo liberal models of efficiency.

AUTHORS’ RESPONSE: Thank you for the comment. We agree that such a suggestion is in place given our findings and strengthens the explicit policy relevance of the manuscript. We have added the following sentence on row 80 of the manuscript:

“Parity of participation, that is, social arrangements that enable all members of a society regardless of status to equally participate in all spheres of social interaction and social life, has been viewed to be at the core of social justice [25]. “

And the following sentence of row 515 of the manuscript:

In contrast, parity of participation in regard to RAS implies a need for policies that ensure that the forced migrant populations’ perspectives are represented, recognized, and valued in all social structures [101]."

We have added the following two references to the list:

Blue, G., M. Rosol, and V. Fast, Justice as Parity of Participation: Enhancing Arnstein’s Ladder Through Fraser’s Justice Framework. Journal of the American Planning Association, 2019. 85(3): p. 363-376.

Dahl, H.M., P. Stoltz, and R. Willig, Recognition, Redistribution and Representation in Capitalist Global Society: An Interview with Nancy Fraser. Acta Sociologica, 2004. 47(4): p. 374-382.

Note that we also realised missing information in the information about funding source in the manuscript. Thus, we have added the following words in row 582: …”and the Swedish Research Council for Health, Working Life and Welfare, Grant Number 2016-07194…”

Reviewer 2 Report

This is an outstanding paper that I look forward to assigning to my students once it is published. 

Beyond the authors' mastery of the topic area, I appreciate seeing that the authors employ a systematic literature review for their research design.

Such approaches are important, and I learned a lot while reading this manuscript.

Author Response

AUTHORS’ RESPONSE: We thank the reviewer for the truly positive comments on the manuscript, and are delighted to hear that it may be used in teaching in the future.

As a response to the reviewer’s recommendation of a minor spell check, we have performed this and corrected a number of minor typos, the corrections of which can be seen in the tracked changes of the resubmitted manuscript.

Note that we also realised missing information in the information about funding source in the manuscript. Thus, we have added the following words in row 582: …”and the Swedish Research Council for Health, Working Life and Welfare, Grant Number 2016-07194…”

Reviewer 3 Report

“A scoping review and conceptual model of social participation and mental health among refugees and  asylum seekers”

Review

The purpose, methods and results of the research are well formulated and structured in the content of the article. There is a large research base that offers increased relevance to research results. The authors present in a transparent manner the limitations of the research, which gives greater credibility to the honesty of the entire research approach. The article presents one of the most complex approaches realized in the specialized literature of the last years regarding the analysis of the social participation of the RAS and the associated aspects of the mental health. I propose the article for publication.

Author Response

AUTHORS’ RESPONSE: We thank the reviewer for the positive comments on the manuscript, and for kindly acknowledging the complexity of the work.

As a response to the reviewer’s recommendation of a minor spell check, we have performed this and corrected a number of minor typos, the corrections of which can be seen in the tracked changes of the resubmitted manuscript.

Note that we also realised missing information in the information about funding source in the manuscript. Thus, we have added the following words in row 582: …”and the Swedish Research Council for Health, Working Life and Welfare, Grant Number 2016-07194…”